# Development of Pedotransfer Functions to Predict Soil Physical Properties in Southern Quebec (Canada)

**Simon Perreault [1]**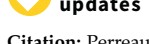**, Anas El Alem [2]**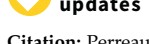**, Karem Chokmani [2],\* and Athyna N. Cambouris [3]**

1. Service de la Planification de L'Aménagement et de L'Environnement, Ville de Québec, Quebec, QC G1K 3G8, Canada; simon.perreault@ville.quebec.qc.ca
2. Centre Eau Terre Environnement, Institut National de la Recherche Scientifique, Quebec, QC G1K 9A9, Canada; anas.el_alem@inrs.ca
3. Agriculture and Agri-Food Canada, Quebec Research and Development Centre, 2560 Hochelaga Boulevard, Quebec, QC G1V 2J3, Canada; athyna.cambouris@agr.gc.ca
\* Correspondence: karem.chokmani@inrs.ca; Tel.: +1-418-654-2570

**Abstract:** Pedotransfer functions (PTFs) are empirical fits to soil property data and have been used as an alternative tool to in situ measurements for estimating soil hydraulic properties for the last few decades. PTFs of Saxton and Rawls, 2006 (PTFs'S&R.2006) are some of the most widely used because of their global aspect. However, empirical functions yield more accurate results when trained locally. This study proposes a set of agricultural PTFs developed for southern Quebec, Canada for three horizons (A, B, and C). Four response variables (bulk density ($\rho_b$), saturated hydraulic conductivity ($K_{sat}$), volumetric water content at field capacity ($\theta_{33}$), and permanent wilting point ($\theta_{1500}$)) and four predictors (clay, silt, organic carbon, and coarse fragment percentages) were used in this modeling process. The new PTFs were trained using the stepwise forward regression (SFR) and canonical correlation analysis (CCA) algorithms. The CCA- and SFR-PTFs were in most cases more accurate. $\Theta_{1500}$ and at $\theta_{33}$ estimates were improved with the SFR. The $\rho_b$ in the A horizon was moderately estimated by the PTFs'S&R.2006, while the CCA- and SFR-PTFs performed equally well for the B and C horizons, yet qualified weak. However, for all PTFs for all horizons, $K_{sat}$ estimates were unacceptable. Estimation of $\rho_b$ and $K_{sat}$ could be improved by considering other morphological predictors (soil structure, drainage information, etc.).

**Keywords:** bulk density; saturated hydraulic conductivity; volumetric water content; modeling; stepwise forward regression; canonical correlation analysis; horizon



## 1. Introduction

A thorough knowledge of soil physical properties is important for crop production, water resource management, erosion risk prevention, contaminant discharge, and flooding interventions. Measurement of soil physical properties such as porosity and saturated hydraulic conductivity can be expensive and time-consuming. In order to avoid laborious measurements, Pedotransfer functions (PTFs) are used as predictors to estimate the physical characteristics of soil by using soil properties that are abundant, easy to measure, and inexpensive. PTFs are frequently developed to estimate volumetric water content for any given matric potential, porosity, saturated hydraulic conductivity, or bulk density. PTFs are also used to estimate plant available water [1,2], to model physical properties of soil during seasonal evapotranspiration [3], or to characterize the parameters of water retention curve models [4,5]. The most common predictors of soil physical properties are soil particle size distribution, organic matter content, coarse fragment content, and sometimes bulk density. Some authors also used texture class [6], moisture class [7], and soil morphological data such as soil structure [8] and color [9].

According to Patil and Singh [10], there exist two methods of PTF development: mechanistic and empirical approaches. *Mechanistic approaches* translate easily measured soil

properties such as texture, bulk density, and particle density into an equivalent pore size distribution model. This model is then related to water content at different soil matric heads. The physico-empirical model of Arya, et al. [11] is one of the most popular mechanistic approaches. *Empirical approaches*, on the other hand, fit a correlation function between the predictor and response variables. Two empirical approaches are commonly used: statistical regressions [12–14] and data mining and exploration techniques [15–18]. Data mining and exploration techniques include, among others, regression trees [19,20], artificial neural networks [21,22], and group methods of data handling [23,24]. The results of empirical approach-based PTFs can take the form of a numeric value or a characteristic class. Most PTFs, however, are developed for a given local or regional pedoclimatic context and are, therefore, site-specific and not universally transferable [25].

As an alternative solution, global datasets have been used in previous studies instead of local or regional datasets, in which case the authors included pedoclimatic predictors such as temperature and moisture [26,27]. For example, the PTFs developed by Saxton and Rawls [28] (referred to here as PTFs'S&R.2006) include soil water characteristic equations formed from the US Department of Agriculture soil database using the available soil texture and organic matter variables. In fact, this is an update of the PTFs developed by Saxton et al. [29], including more variables and a wider range of application. They have been combined with previously reported relationships for stresses and conductivities and the effects of density, gravel, and salinity to form a comprehensive system for predicting soil water characteristics for agricultural water management and hydrologic analyses. Hence, they are popular and commonly used in soil microclimate modeling [30,31]. PTFs'S&R.2006 are very useful. However, since PTFs are empirical-based algorithms, improved modeling of pedoclimatic predictors could be achieved by using locally trained PTFs [32].

In southern Quebec, PTFs have already been developed to predict organic carbon accumulation in the forest zone [33]. However, no PTFs are currently designed for the agricultural area of southern Quebec. The aim of our study was to develop a new set of PTFs (bulk density, saturated hydraulic conductivity, and volumetric water content measured at two matric potentials: $-33$ kPa (field capacity) and $-1500$ kPa (permanent wilting point)) that are well adapted to the pedoclimatic conditions of the agricultural area of southern Quebec. Two statistical methods were tested for deriving PTFs: stepwise forward regression and canonical correlation analysis. The estimation efficiency of this new set of PTFs was then compared with the existing PTFs developed by Saxton and Rawls (2006). Accuracy was assessed using the cross-validation technique from which the $R^2$, Nash–Sutcliffe efficiency (NSE) index, root-mean-square error (RMSE), and bias were generated.

## 2. Materials and Methods

### 2.1. Study Area

The study was conducted in the Monteregie agricultural area, located southeast of Montréal, Quebec, Canada. The climate of this region is temperate, with an average air temperature of $-10.2\ ^\circ$C in January and $20.4\ ^\circ$C in July [34]. In terms of yearly averages, the duration of the frost-free period is 206.5 days, total rainfall is 931.7 mm, and total snowfall is 224.5 cm. Monteregie is one of the largest and most productive agricultural areas in Quebec [35]. Analyses conducted on both ground and surface water using bacteriological and physicochemical indices revealed that water quality is poor at many sampling points [36]. Soil quality was also affected by nutrient leaching, erosion, and overfertilization [37]. Greater use of beneficial management practices is, therefore, needed to ensure soil and water conservation in this region. Development of a set of appropriate PTFs to estimate secondary soil properties would be helpful for planning beneficial management practices implementation.

A high degree of pedodiversity and soil texture variability is perceived in the study area (Figure 1). The soils in this region are gravelly, sandy, loamy, clayey, and organic soils [38]. Many soil taxonomic orders (as defined by the US system of soil taxonomy)

are present, including spodosols, inceptisols, and histosols [39]. The soils of the region tend to have poor natural drainage; however, after artificial drainage, they become mostly moderately well drained [40]. Several soil surveys of southern Quebec have been updated since 1975 and are available on the Canadian Soil Information Service website [41].

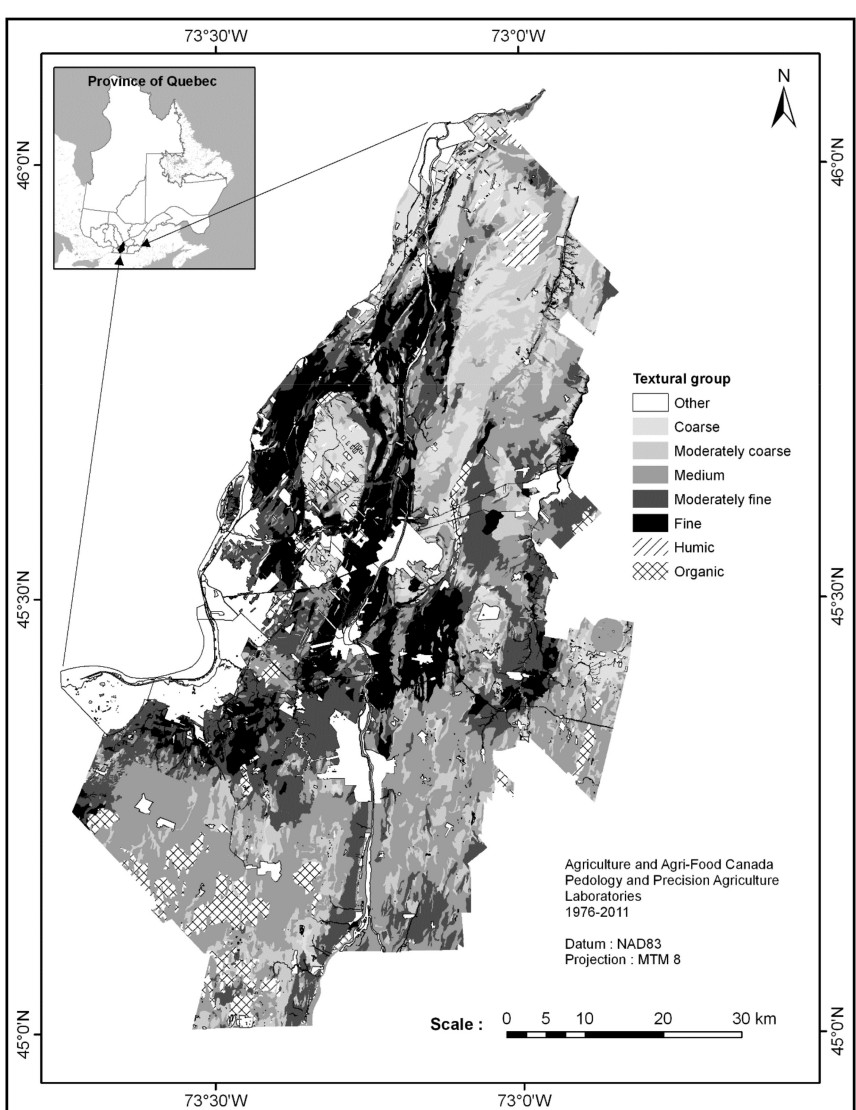

**Figure 1.** Map of the Monteregie soil surface textural groups.

### 2.2. Soil Data

Agriculture and Agri-Food Canada (AAFC) has maintained an analytical soil database for southern Quebec since 1975. This database contains a set of georeferenced samples from A, B, and C horizons. Soil physical data (primary and secondary properties) from the analytical soil database were used to develop and assess the proposed PTFs.

Primary soil properties are the first and second components of particle size distribution, such as clay, silt, organic carbon (OC), and coarse fragment (CF) percentages. These properties, which define soil pore space, have an impact on soil water-holding properties, hydraulic conductivity, and soil bulk density. That is why they were chosen as PTF predictors. To avoid multicollinearity problems, sand percentage was not considered. This choice implies the exclusion of organic soils, since they have no mineral content. Soil texture (clay and silt) was determined by the hydrometer method [42], CF (>2 mm) content was determined by sieving [42], and OC content was determined by the Walkley–Black method [43].

Selected secondary soil properties—bulk density ($\rho_b$), saturated hydraulic conductivity ($K_{sat}$), and volumetric water content ($\theta$) measured at two matric potentials, $-33$ kPa (field capacity ($\theta_{33}$)) and $-1500$ kPa (permanent wilting point ($\theta_{1500}$))—were considered as response variables in the PTFs. The procedures used to measure $\theta$, $\rho_b$, and $K_{sat}$ were described by Topp et al. [44], Culley [45], and Reynolds [46], respectively. The number of samples available for $\theta_{33}$, $\theta_{1500}$, $\rho_b$, and $K_{sat}$ was 328, 327, 352, and 310, respectively. These numbers exclude soils that were entirely defined by primary properties and organic soils.

*2.3. Statistical Methods*

For a given soil profile, the distribution of soil properties changes with depth. For instance, OC tends to decrease with increasing depth [47]. In this paper, most of the studied soils are tilled, which can also influence physical properties ($\rho_b$, $\theta$, and $K_{sat}$) of the A horizon [48]. Consequently, a PTF solely based on A horizon properties is not suitable to estimate soil physical properties at other depths. The dataset used for PTF development in this paper was stratified according to soil horizons (A, B, and C).

Before developing new PTFs, a preliminary study of predictors and response variables is essential [49]. This preliminary study was conducted on each soil dataset corresponding to a soil horizon. The first step was outlier detection. Values larger than three standard deviations from the mean value were regarded outliers and removed from the dataset. Two development approaches were tested: one based on stepwise forward regression (SFR) and the other based on canonical correlation analysis (CCA). The CCA method requires that each soil sample contains all four predictors (clay, silt, OC, and CF) along with the four response variables ($\theta_{1500}$, $\theta_{33}$, $\rho_b$, and $K_{sat}$). To be consistent in both CCA and SFR development methods, only the soil samples having these four predictors and four response variables were kept. In statistical regressions, predictors and response variables must be normally distributed. In order to respect the normality hypothesis, some variables were transformed using the Box–Cox algorithm. The Box–Cox applies a power transformation $\lambda$, which maximizes a log-likelihood function (Equation (1)). When the value of $\lambda$ is 1, a logarithm transformation is applied (Equation (2)).

$$\frac{x^{\lambda} - 1}{\lambda}, \lambda \neq 0. \tag{1}$$

$$\ln(x), \lambda = 1. \tag{2}$$

A correlation study was performed on both predictors and response variables. A strong correlation between predictors indicates that the information is redundant. Adding highly correlated predictors to a PTF will not improve its prediction potential. By contrast, a strong correlation between a predictor and a response variable indicates that selection of the predictor will improve the PTF. Correlation between response variables indicates the linearity of these secondary soil properties.

2.3.1. Stepwise Forward Regression (SFR)

SFR is a commonly used method in statistical regression to identify the most significant predictors when estimating a response variable. The selection of predictors is based on entrance and exit thresholds, which are set according to the *p*-value of regression coefficients. The *p*-value is computed to decide whether a given input variable should be considered as a predictor or not. This *p*-value must be lower than the entrance significance threshold; otherwise, the predictor is rejected. Each time a new predictor is accepted in the regression, the *p*-values of all previously accepted predictors are recomputed; predictors are retained only if their new values are lower than the exit threshold. Significance levels are fixed at 5% for entrance and exit from the regression model. The method is applied to each response variable using the training dataset.

2.3.2. Canonical Correlation Analysis (CCA)

The objective of a CCA is to transform predictor ($x$) and response ($y$) variables using linear combinations in two sets of canonical variables, $U_j$ and $V_j$ respectively. The parameters are calculated such that the correlation between canonical variables $U_j$ and $V_j$ is maximal. However, the internal correlation of the different components of each canonical variable is minimal. Canonical variables are generated using canonical coefficients ($a_{ij}$ and $b_{ij}$) and mean-centered variables (see Equations (3) and (4)).

$$U_j = \sum_{i,j=1}^{N} (x_i - \overline{x_i})a_{ij}, \tag{3}$$

$$V_j = \sum_{i,j=1}^{N} (y_i - \overline{y_i})b_{ij}, \tag{4}$$

where $x_i$ is an observed value of a predictor, $\overline{x_i}$ is the average value of the predictor, $y_i$ is an observed value of a response variable, and $\overline{y_i}$ is the average value of the response variable.

One of the attractive features of CCA is that canonical variables enable grouping redundant properties into a single component. Each canonical variable related to predictors ($U_j$) contains the maximum amount of information available having an optimal correlation with the canonical variable related to the response variables ($V_j$) [50]. However, in the context of the PTF development, the goal is not to generate canonical response variables $V_j$, but rather to predict response variables. Nevertheless, if there are interconnections between secondary and primary soil properties, it can be useful to perform a CCA with predictor and response variables and a multiple regression using $U_j$ as predictors of the response variables. In doing so, a maximum of information derived from the predictors is translated into $U_j$ canonical variables, which are then optimized according to the $V_j$ canonical variables computed from the response variables. It has been demonstrated that performing a regression on a system of canonical variables gives satisfactory results when used in an estimation process in the presence of collinearity [51].

The CCA-PTF development method is depicted in Figure 2. The first step of this method is to conduct a CCA with predictors $x_i$ (clay, silt, OC, and CF) and response variables $y_i$ ($\theta_{1500}$, $\theta_{33}$, $\rho_b$, and $K_{sat}$) according to a training dataset for a given soil horizon. The next step is to keep the $U_j$ canonical variables and their canonical coefficients $a_{ij}$. $V_j$ canonical variables and their canonical coefficients $b_{ij}$ are not used. As mentioned before, each $U_j$ has a maximum correlation with its corresponding $V_j$ calculated from the response variables, $y_i$. A multiple regression is then performed for each of the response variables and $U_j$ variables from the training dataset. Only $U_j$ variables with a significant regression coefficient are retained in the regression (the coefficient must differ from 0 with a significance $p$-value of 5%). Regressions are then recomputed for all other response variables from the training dataset. The next step is to produce a set of canonical variables using predictors $x_i$ from a validation dataset, as well as the previously generated mean predictors and canonical coefficients $a_{ij}$. Regression parameters and previously determined intercepts (from the training dataset) are then used with the new canonical variables to estimate the response variable $y_i$. Finally, accuracy of the results is assessed by comparing the estimated response variables and response variables from the validation dataset.

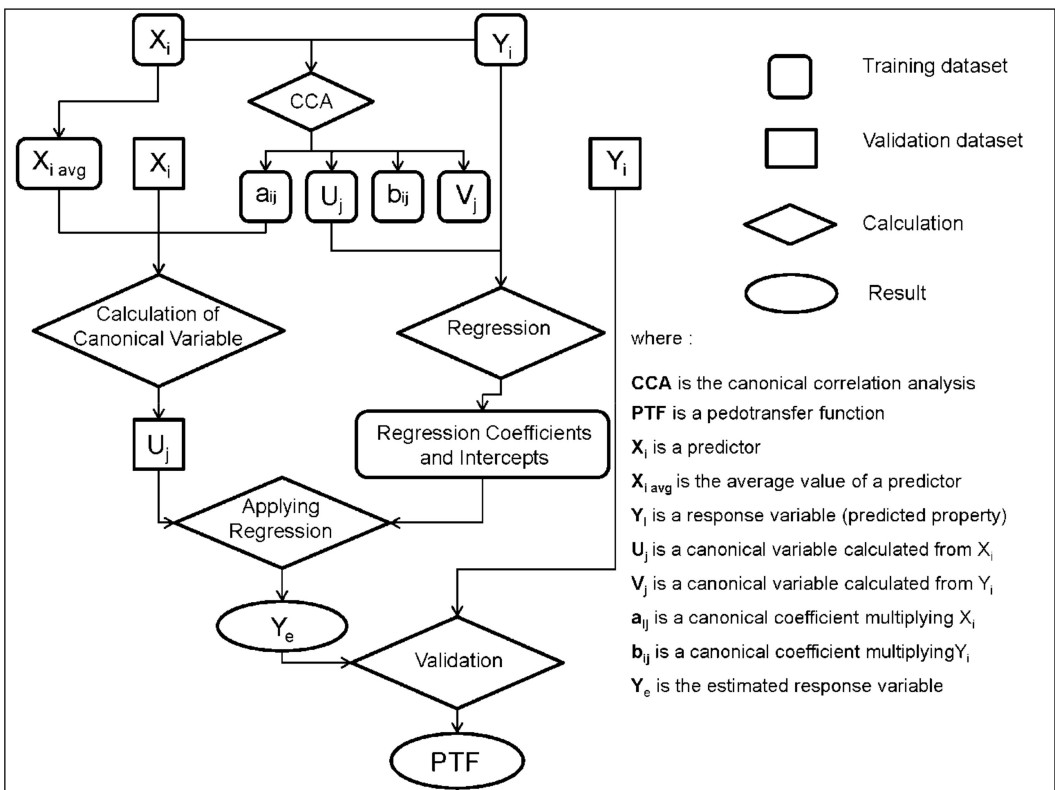

**Figure 2.** Development procedures of PTFs using CCA method.

### 2.3.3. Accuracy Assessment

To evaluate the estimation quality of the developed PTFs, the determination coefficient ($R^2$), root-mean-square error (RMSE), bias, and Nash–Sutcliffe efficiency (NSE) index [52] were calculated for each secondary soil property during the accuracy assessment procedure. The RMSE (Equation (5)) quantifies the contribution of systematic and random errors expressed in measurement units, and the bias (Equation (6)) quantifies the systematic error (over- or underestimation), also expressed in measurement units [53]. The NSE (Equation (7)) is used to characterize the goodness of fit of a model. NSE values can range from $-\infty$ to 1. An NSE value equal to 1 indicates perfect modeling, while a value below 0 means that the average of measured values is a better predictor than the model predictions. When the NSE value is equal to 0, the performance of both predictors is similar. A classification of the NSE applied to the PTFs is detailed in Table 1.

**Table 1.** Classification of the Nash–Sutcliffe efficiency (NSE) index.

| NSE Value | Qualification |
|:---:|:---:|
| $\leq 0$ | Unacceptable |
| 0 to 0.4 | Weak (unsatisfactory) |
| 0.4 to 0.6 | Moderate (satisfactory) |
| 0.6 to 0.8 | Good (satisfactory) |
| 0.8 to 1 | Optimal (satisfactory) |

A cross-validation procedure was conducted on the developed PTFs. Each horizon dataset was randomly split into two datasets: one for training and one for validation [54]. A ratio of 1: 4 was used to randomly split each horizon dataset into a training dataset (75%) and a validation dataset (25%). Minimum and maximum values for each response variable were assigned to the training dataset to prevent extrapolation. This procedure was repeated 10,000 times from the original horizon dataset. At each loop, a PTF was calculated from the training dataset and then applied on the validation dataset. Each metric ($R^2$, RMSE, bias,

and NSE) was calculated from the estimated and the measured soil physical property of the validation dataset. At the end of the final iteration, the average and confidence intervals of each metric were calculated (probability = 95%). The existing PTFs were also evaluated at each horizon using the corresponding dataset. The same metrics were calculated but in a context of independent validation.

$$RMSE = \sqrt{\frac{1}{N}\sum_{i=1}^{N}(E_i - M_i)^2}, \tag{5}$$

$$Bias = \frac{1}{N}\sum_{i=1}^{N}(E_i - M_i), \tag{6}$$

$$NSE = 1 - \frac{\sum_{i=1}^{N}(E_i - M_i)^2}{\sum(M_i - M)}, \tag{7}$$

where $E$ is the estimated value, $M$ is the measured value, and $N$ is the number of samples.

## 3. Results

### 3.1. Data Exploration

Means and coefficients of variation (CV) were calculated for each soil parameter and horizon dataset (Table 2). The mean percentage of OC, $\theta_{1500}$, and $K_{sat}$ decreased with soil depth (from the A to the C horizon), while $\rho_b$ increased from the A to the C horizon. OC is the result of plant decomposition and manure application, which take place within the top 20 cm soil layer. Soil compaction increases with depth $\rho_b$ and reduces the porosity, thus determining the available space for water. It should be noted that CVs were generally lower than 1, except for CF and $K_{sat}$, which often show very high spatial variability at the field scale [55].

**Table 2.** Statistics of soil properties; CV, coefficient of variation.

| Soil Properties | | A Horizon | | B Horizon | | C Horizon | |
|---|---|---|---|---|---|---|---|
| | | Mean | CV | Mean | CV | Mean | CV |
| Primary | Silt (%) | 35.14 | 0.42 | 33.86 | 0.52 | 35.10 | 0.45 |
| | Clay (%) | 21.82 | 0.62 | 22.23 | 0.89 | 18.89 | 0.94 |
| | OC (%) | 2.21 | 0.49 | 0.46 | 1.02 | 0.16 | 0.81 |
| | CF (%) | 3.62 | 1.81 | 7.85 | 1.63 | 7.79 | 1.63 |
| Secondary | $\theta_{33}$ (%) | 31.66 | 0.20 | 28.36 | 0.37 | 29.25 | 0.41 |
| | $\theta_{1500}$ (%) | 15.48 | 0.38 | 14.18 | 0.62 | 13.30 | 0.69 |
| | $\rho_b$ (g·cm$^{-3}$) | 1.33 | 0.12 | 1.50 | 0.14 | 1.60 | 0.15 |
| | $K_{sat}$ (cm·h$^{-1}$) | 22.30 | 1.23 | 14.95 | 1.59 | 5.56 | 2.51 |
| | Sample size | 88 | | 121 | | 95 | |

Statistical analysis of soil parameters was conducted on normally distributed data. The data transformations applied to each variable (when required) are shown in Table 3. Distributions, scatter plots, and Pearson's correlation coefficients of both primary and secondary soil properties are illustrated in a matrix form for each soil horizon in Figure 3. The cross-correlation coefficients obtained between predictors were below 0.5, absolute value, at all soil horizons. However, these correlation coefficients were often significant, which means that certain predictors were collinear (i.e., silt, clay, etc.).

**Table 3.** Normal distribution transformations of soil properties.

| Soil Properties | | A Horizon | B Horizon | C Horizon |
|---|---|---|---|---|
| Primary | Silt | – | – | – |
| | Clay | Box–Cox, $\lambda = 0.4072$ | Box–Cox, $\lambda = 0.1248$ | Box–Cox, $\lambda = 0.1659$ |
| | OC | Box–Cox, $\lambda = 0.2750$ | Box–Cox, $\lambda = -0.0874$ | Box–Cox, $\lambda = 0.1664$ |
| | CF | $\ln(x + 1)$ | $\ln(x + 1)$ | $\ln(x + 1)$ |
| Secondary | $\theta_{33}$ | – | – | – |
| | $\theta_{1500}$ | – | Box–Cox, $\lambda = 0.2258$ | Box–Cox, $\lambda = 0.2206$ |
| | $\rho_b$ | – | Box–Cox, $\lambda = 2.4488$ | Box–Cox, $\lambda = 2.2516$ |
| | $K_{sat}$ | $\ln(x + 1)$ | $\ln(x + 1)$ | $\ln(x + 1)$ |

A horizon

B horizon

C horizon

**Figure 3.** Cross-correlation and distribution matrix of primary and secondary soil properties. [1] Transformed data; * significant correlation ($p = 0.05$).

Most of the highest correlation coefficients, in absolute values, were obtained between predictors and response variables. $\theta_{33}$ was positively correlated with silt (0.60, 0.49, and 0.50) and clay (0.52, 0.80, and 0.80), at the A, B, and C horizons, respectively. $\theta_{1500}$ was also positively correlated with silt (0.60, 0.46, and 0.51) and clay (0.76, 0.86, and 0.89), at the A, B, and C horizons, respectively. Clay content has an impact on water retention properties, as water content tends to be greater in soils that have a high clay percentage. A negative correlation was observed between $\rho_b$ and OC percentage (−0.56, −0.52, and −0.55) at the A, B, and C horizons, respectively. In the C horizon, clay percentage was negatively correlated to $\rho_b$ (−0.52). $K_{sat}$ had weak correlation coefficients with all predictors for the A horizon. However, negative correlations of −0.44 and −0.56 with silt percentage were observed at the B and C horizons, respectively. The effect of CF on $K_{sat}$ can be either negative or positive, depending on CF location. When CFs are on the soil surface, they increase infiltration by preventing the soil from sealing, but they reduce infiltration when contained within the soil layer [56–58]. The relationship between OC and $K_{sat}$ can also vary.

Some authors argue that OC increases $K_{sat}$ by improving soil structure [59], while others conclude that it decreases $K_{sat}$ because OC retains water, its aggregates increase tortuosity, and the quality/kind of organic matter may affect hydraulic properties [23].

### 3.2. Stepwise Forward Regression-Based Pedotransfer Functions

Table 4 shows the regression coefficients obtained using the SFR method. Standardized coefficients show the weight accorded to predictors in the developed PTF (coefficients without unit effect measurement).

**Table 4.** Standardized (St. *) and non-standardized (Non st. **) regression coefficients obtained by stepwise forward regression.

| Response Variables | $\theta_{33}$ (%) | | $\theta_{1500}$ (%) | | $\rho_b$ (g cm$^{-3}$) | | $K_{sat}$ (cm h$^{-1}$) | |
|---|---|---|---|---|---|---|---|---|
| Horizon | Coefficients | | Coefficients | | Coefficients | | Coefficients | |
| | St. * | Non st. ** | St. * | Non st. ** | St. * | Non st. ** | St.* | Non st. ** |
| **A** | | | | | | | | |
| Intercept | – | 18.3997 | – | – | – | 1.5525 | – | 1.7828 |
| Silt | 2.7664 | 0.1903 | 1.8400 | 0.1397 | – | – | – | – |
| Clay [1] | 1.3949 | 0.6302 | 3.1261 | 1.5100 | −0.0431 | −0.0195 | – | – |
| OC [1] | 3.1786 | 4.1219 | 1.6514 | 2.2206 | −0.1250 | −0.1623 | 0.3497 | 0.4543 |
| CF [1] | – | – | – | – | – | – | 0.4054 | 0.3674 |
| $R^2$ | | 0.59 | | 0.69 | | 0.50 | | 0.2694 |
| RMSE | | 4.34 | | 3.3012 | | 0.14 | | 42.87 |
| **B** | | | | | | | | |
| Intercept | – | 10.6471 | – | 0.7646 | – | 0.6123 | – | 3.4065 |
| Silt | 1.6178 | 0.0943 | 0.1440 | 0.0084 | 0.1157 | 0.0067 | −0.58 | −0.0340 |
| Clay [1] | 7.2655 | 5.4649 | 0.9087 | 0.6835 | −0.1494 | −0.1123 | – | – |
| OC [1] | 1.2338 | 1.1441 | | – | −0.2076 | −0.1925 | 0.6102 | 0.5658 |
| CF [1] | −2.3443 | −1.6381 | | – | – | – | 0.3696 | 0.2583 |
| $R^2$ | | 0.70 | | 0.73 | | 0.49 | | 0.28 |
| RMSE | | 6.03 | | 4.73 | | 0.18 | | 29.035 |
| **C** | | | | | | | | |
| Intercept | – | 21.2177 | – | 0.6406 | – | −0.0217 | – | 3.7264 |
| Silt | 2.8625 | 0.1815 | 0.1788 | 0.0113 | 0.1198 | 0.0076 | −0.5898 | −0.0374 |
| Clay [1] | 6.3755 | 4.0055 | 1.0586 | 0.6651 | −0.1188 | −0.0747 | −0.2486 | −0.1562 |
| OC [1] | 2.3907 | 4.1961 | – | – | −0.2278 | −0.3998 | 0.2729 | 0.4790 |
| CF [1] | −3.5305 | −2.6145 | – | – | 0.1444 | 0.1069 | – | – |
| $R^2$ | | 0.68 | | 0.74 | | 0.52 | | 0.23 |
| RMSE | | 7.39 | | 4.78 | | 0.21 | | 25.92 |

For *the estimation of $\theta_{33}$*, the A horizon PTF used the following predictors: OC, silt, and clay percentages, listed by decreasing weight. The PTFs developed for the B and C horizons had the highest weight of clay, followed by CF percentage, and small weights of silt and OC. The standardized regression coefficients of OC were stronger for the A horizon, which is explained by the abundance of OC on the soil surface.

In the case of *the estimation of $\theta_{1500}$*, clay and silt, followed by OC, were retained as predictors for the A horizon. Coefficient values were positive, indicating that increases in these properties correspond to increases in $\theta_{1500}$. At the B and C horizons, clay percentage coefficients were broadly higher, followed by silt percentage. As shown in Figure 3, the correlation between $\theta_{1500}$ and clay was the highest at each soil horizon. Increasing clay content increases the soil water-holding capacity.

In the case of *the estimation of $\rho_b$*, a negative weight was given to OC and clay content for all horizons. As mentioned above, a negative correlation was observed between $\rho_b$ and

these predictors. At the B horizon a moderate positive weight was also given to silt content. At the C horizon, moderate positive weights were given to CF and silt percentages.

The predictors selected for *the estimation of $K_{sat}$* were different for each horizon (Table 4). At the A horizon, only CF and OC were selected with a positive weight. OC was also given a positive weight at the B horizon, followed by silt with a negative weight, and CF with a positive weight. At the C horizon, silt was selected with a negative weight, followed by CF (positive weight) and OC (negative weight).

### 3.3. Canonical Correlation Analysis-Based Pedotransfer Functions

Correlations between canonical variables U and V are presented in Table 5. As expected, the correlation decreased from the first to the fourth canonical variable. The weights accorded by the PTF to a predictor were determined using a combination of the weight of the predictors related to the canonical variables (Table 6) and the weight given to the canonical variables obtained by regression (Table 7) between response variables. Because of the scale effect, it is recommended to use correlation coefficients to describe the contribution of a predictor to a canonical variable, instead of interpreting canonical coefficients $a_{ij}$ [60]. Canonical coefficients are only used to calculate canonical variables.

**Table 5.** Correlation coefficients between canonical variables U and V calculated for each soil horizon.

|  | A Horizon | B Horizon | C Horizon |
|---|---|---|---|
| $U_1,V_1$ | 0.89 | 0.89 | 0.90 |
| $U_2,V_2$ | 0.52 | 0.69 | 0.66 |
| $U_3,V_3$ | 0.43 | 0.28 | 0.40 |
| $U_4,V_4$ | 0.26 | 0.07 | 0.01 |

**Table 6.** Canonical coefficients $a_{ij}$ and correlation coefficients generated for each canonical variable.

| Property | $U_1$ | | $U_2$ | | $U_3$ | | $U_4$ | |
|---|---|---|---|---|---|---|---|---|
|  | $a_{ij}$ | $R$ | $a_{ij}$ | $R$ | $a_{ij}$ | $R$ | $a_{ij}$ | $R$ |
| **Horizon A** | | | | | | | | |
| Silt | 0.0217 | 0.61 | 0.0118 | 0.29 | 0.0382 | 0.52 | −0.0620 | −0.52 |
| Clay [1] | 0.2133 | 0.74 | 0.3311 | 0.59 | −0.1730 | −0.10 | 0.2647 | 0.31 |
| OC [1] | 0.7723 | 0.74 | −1.0277 | −0.60 | −0.3633 | −0.29 | −0.0224 | −0.04 |
| CF [1] | −0.0673 | −0.20 | 0.2768 | 0.14 | −0.6684 | −0.80 | −0.5843 | −0.54 |
| $\theta_{33}$ | – | 0.75 | – | −0.06 | – | 0.10 | – | −0.12 |
| $\theta_{1500}$ [1] | – | 0.79 | – | 0.23 | – | 0.03 | – | −0.01 |
| $\rho_b$ [1] | – | −0.63 | – | 0.27 | – | 0.14 | – | −0.09 |
| $K_{sat}$ [1] | – | 0.13 | – | −0.07 | – | −0.42 | – | 0.00 |
| **Horizon B** | | | | | | | | |
| Silt | −0.0101 | −0.58 | −0.0329 | −0.45 | −0.0066 | −0.07 | −0.0551 | −0.68 |
| Clay [1] | −0.6223 | −0.97 | 0.1766 | 0.10 | −0.4248 | −0.16 | 0.4601 | 0.14 |
| OC [1] | −0.0625 | −0.24 | 0.7298 | 0.83 | 0.4645 | 0.18 | −0.4714 | −0.47 |
| CF [1] | 0.1170 | 0.47 | 0.1326 | 0.38 | −0.7419 | −0.77 | −0.1152 | −0.22 |
| $\theta_{33}$ | – | −0.84 | – | −0.01 | – | 0.07 | – | −0.02 |
| $\theta_{1500}$ [1] | – | −0.81 | – | 0.03 | – | −0.09 | – | 0.0185 |
| $\rho_b$ [1] | – | 0.32 | – | −0.59 | – | −0.10 | – | −0.01 |
| $K_{sat}$ [1] | – | 0.30 | – | 0.60 | – | −0.05 | – | −0.02 |
| **Horizon C** | | | | | | | | |
| Silt | 0.0103 | 0.58 | 0.0458 | 0.65 | −0.0372 | −0.31 | −0.0428 | −0.37 |
| Clay [1] | 0.4811 | 0.97 | 0.0788 | 0.04 | 0.5646 | 0.20 | 0.4199 | 0.10 |
| OC [1] | 0.3016 | 0.54 | −1.3032 | −0.62 | −0.4637 | 0.05 | −1.4239 | −0.57 |
| CF [1] | −0.1148 | −0.43 | 0.2432 | 0.20 | 0.6708 | 0.72 | −0.3155 | −0.51 |
| $\theta_{33}$ | – | 0.81 | – | 0.00 | – | −0.15 | – | 0.00 |
| $\theta_{1500}$ [1] | – | 0.88 | – | 0.07 | – | 0.08 | – | 0.00 |
| $\rho_b$ [1] | – | −0.50 | – | 0.53 | – | 0.07 | – | 0.00 |
| $K_{sat}$ [1] | – | −0.39 | – | −0.46 | – | 0.19 | – | 0.00 |

**Table 7.** Regression coefficients between canonical variables and response variables; $p \leq 0.05$.

| Response Variables | | | | |
|---|---|---|---|---|
| **Horizon A** | $\theta_{33}$ | $\theta_{1500}$ [1] | $\rho_b$ [1] | $K_{sat}$ [1] |
| Intercept | 32.4341 | 15.7784 | 1.2982 | 2.4895 |
| $U_1$ | 5.1244 | 4.6768 | −0.1266 | – |
| $U_2$ | – | 1.3632 | 0.0546 | – |
| $U_3$ | – | – | – | −0.5768 |
| $U_4$ | – | – | – | – |
| $R^2$ | 0.57 | 0.69 | 0.48 | 0.19 |
| RMSE | 4.47 | 3.28 | 0.14 | 43.40 |
| **Horizon B** | $\theta_{33}$ | $\theta_{1500}$ [1] | $\rho_b$ [1] | $K_{sat}$ [1] |
| Intercept | 28.7736 | 3.3318 | 0.7082 | 1.8381 |
| $U_1$ | −9.4011 | −0.9740 | 0.1305 | 0.4429 |
| $U_2$ | – | – | −0.2357 | 0.8850 |
| $U_3$ | – | – | – | – |
| $U_4$ | – | – | – | – |
| $R^2$ | 0.70 | 0.71 | 0.47 | 0.23 |
| RMSE | 6.08 | 4.93 | 0.18 | 29.55 |
| **Horizon C** | $\theta_{33}$ | $\theta_{1500}$ [1] | $\rho_b$ [1] | $K_{sat}$ [1] |
| Intercept | 29.5832 | 3.1433 | 0.8339 | 1.1059 |
| $U_1$ | 10.6868 | 1.1536 | −0.2313 | −0.4499 |
| $U_2$ | – | – | 0.2428 | −0.5372 |
| $U_3$ | – | – | – | 0.2169 |
| $U_4$ | – | – | – | – |
| $R^2$ | 0.66 | 0.73 | 0.50 | 0.27 |
| RMSE | 7.66 | 4.89 | 0.21 | 25.41 |

### 3.3.1. Contribution of Predictors to Canonical Variables

At the A horizon, the predictors making the highest contribution to $U_1$ were OC and clay percentages, followed by silt with a moderate correlation coefficient (Table 6). The main contributions to $U_1$ at the B and C horizons came from the clay percentage followed by the silt percentage and OC. The OC contribution to the B and C horizons was weak compared to its contribution to the A horizon. The dominant contributions to $U_2$ at the A horizon (highest negative correlation) came from OC and clay content, with a similar contribution but with positive value (Table 6). OC was also the predictor with the highest correlation at the B horizon, but with a slightly lower and positive value. Silt percentage made a moderate negative contribution. At the C horizon, the contributor with the highest correlation was silt, followed by OC, with a similar absolute contribution. Thus, $U_2$ is essentially explained by the OC contribution, although, depending on the horizon, clay and silt contribute to that canonical variable. For all soil horizons, the predictor with the highest correlation with $U_3$ was CF (Table 6). It should be noted that a negative correlation was observed with both the A and B horizons. It was not possible, regarding $U_4$, to identify one or two predictors that were applicable to all three horizons. In terms of predictive power, the first canonical variable $U_1$, showed the highest correlation with most response variables, in all horizons. The strength of correlation then decreased from $U_2$ to $U_4$ with most response variables.

### 3.3.2. Regressions Using Canonical Variables as Predictors

For the *estimation of* $\theta_{33}$, $U_1$ was the only canonical variable selected for all horizons, with correlation coefficients varying between 0.75 and 0.84 (Table 6). $U_1$ was essentially defined by clay and silt percentages in these horizons, but OC also made a strong contribution in the A horizon. The weights given to the predictors were similar to those obtained with the SFR approach. The results of the regression used to *estimate* $\theta_{1500}$ were similar to those estimating $\theta_{33}$, in terms of predictor weight (Table 6) and regression coefficients of

the canonical variables (Table 7). A strong correlation between $\theta_{1500}$ and $\theta_{33}$ was previously observed (Figure 3). $U_1$ was strongly correlated with $\theta_{1500}$ (Table 6), which explains why it was once again retained in the regression (Table 7). Clay also had a large impact on these functions. $U_2$ was selected for the A horizon, which was mostly correlated with clay and OC contents. Again, the resulting PTF was similar to the one developed with the SFR approach.

For each horizon, $U_1$ and $U_2$ were selected to *estimate* $\rho_b$ (Table 7). As previously mentioned, $U_1$ was explained by clay, silt, and OC percentages, while $U_2$ was mostly explained by OC, followed by silt percentage for B and C horizons. OC was more important in the A horizon and decreased with increasing soil depth. Multiplication of the correlation coefficient (Table 6) by the regression coefficients (Table 7) gives the effect of a predictor on a response variable. Negative weights were given to clay and OC percentages. Silt had a negative effect at the A horizon and a positive effect at the B and C horizons. Organic matter has a lower $\rho_b$ than mineral material; thus, the overall $\rho_b$ is reduced when the organic matter percentage in mineral soil increases. This could explain the negative effect of OC in both CCA- and SFR-derived PTFs for $\rho_b$ prediction. The negative contribution of clay in PTFs predicting $\rho_b$ was also observed by Jones [61].

In the case of $K_{sat}$ *estimation*, the correlation with canonical variables was different for each horizon. For the A horizon, $K_{sat}$ was slightly correlated to $U_3$ (negatively; Table 6). The regression between $K_{sat}$ and its canonical variables gave a negative weight to $U_3$ (Table 7). CF was the main predictor for $U_3$ (negative weight), followed by silt (positive weight) and OC (negative weight). The resulting combination of correlation and weight showed the positive effects of OC and CF, and the negative effect of silt on $K_{sat}$. These results are consistent with the previously conducted cross-correlation analysis (Figure 3). For the B horizon, the regression selected $U_2$ and $U_1$ with positive weights (Table 7). $U_2$ was correlated with $K_{sat}$, followed by $U_1$ (moderately). As a result, OC was the most contributing predictor (positive effect), followed by silt and clay (negative), and CF (small positive). These results are consistent with those of the cross-correlation analysis (Figure 3), where $K_{sat}$ was positively correlated with OC and CF, and negatively correlated with silt and clay percentages. For the C horizon, $U_3$ was selected as a predictor with positive weight, while $U_2$ and $U_1$ were selected with negative weights. The correlation between $U_1$ and $K_{sat}$ was similar to that obtained with $U_2$ in absolute terms. However, the correlation obtained with $U_3$ was weak (Table 6). The negative coefficients for $U_1$ and $U_2$ indicate that clay and silt made a negative contribution to this PTF, while the effect was negative for OC. The positive correlation coefficient for $U_3$ was explained by a positive effect of CF.

Neither the SFR nor the CCA method selected identical predictors at each horizon. However, it is well known that higher clay-to-silt percentages increase soil water retention and that higher CF reduces this property [62]. In this study, increased OC content led to increased $K_{sat}$. However, it has been demonstrated that the effect of clay is positive when its proportion is less than 30%, but varied and more complex when the proportion is higher [63].

*3.4. Validation and Comparison with the Saxton and Rawls's PTFs*

Table 8 presents accuracy assessment results of the PTFs ($R^2$, RMSE, NSE index, and bias) for both developed methods (SFR and CCA) and for Saxton and Rawls's PTFs (2006), which are further referred to as PTFs'S&R.2006.

**Table 8.** Accuracy evaluation of existing and developed PTFs. Values in bold are the best performances. The values in square brackets are the maximum and minimum values calculated from the simulations.

| Property/Horizon | Saxton & Rawls * PTFs | | | | Stepwise Forward Regression | | | | Regression with CCA | | | |
|---|---|---|---|---|---|---|---|---|---|---|---|---|
| | $R^2$ | NSE | RMSE | Bias | $R^2$ | NSE | RMSE | Bias | $R^2$ | NSE | RMSE | Bias |
| $\theta_{33}$ (%) | | | | | | | | | | | | |
| A | 0.45 | −0.26 | 7.6 | −3.4 | **0.54** [0.540, 0.544] | **0.47** [0.466, 0.470] | **4.3** [4.29, 4.31] | **0.4** [0.41, 0.45] | 0.53 [0.525, 0.529] | **0.47** [0.467, 0.471] | 4.4 [4.36, 4.38] | **0.4** [0.34, 0.38] |
| B | 0.74 | 0.54 | 7.5 | −3.5 | 0.68 [0.675, 0.678] | 0.63 [0.626, 0.629] | 5.9 [5.91, 5.93] | 0.8 [0.79, 0.83] | 0.68 [0.679, 0.682] | 0.63 [0.630, 0.633] | 5.9 [5.86, 5.88] | 0.8 [0.78, 0.82] |
| C | 0.66 | 0.34 | 10.7 | −7.2 | 0.69 [0.691, 0.693] | 0.65 [0.652, 0.655] | 7.4 [7.34, 7.37] | **0.5** [0.42, 0.48] | **0.70** [0.697, 0.699] | **0.66** [0.660, 0.663] | **7.3** [7.28, 7.31] | **0.5** [0.51, 0.56] |
| $\theta_{1500}$ (%) | | | | | | | | | | | | |
| A | 0.56 | 0.32 | 4.8 | −0.2 | **0.68** [0.677, 0.680] | **0.67** [0.671, 0.675] | **3.2** [3.19, 3.21] | **0.2** [0.24, 0.26] | 0.66 [0.659, 0.662] | 0.61 [0.609, 0.613] | 3.3 [3.32, 3.34] | 0.3 [0.25, 0.28] |
| B | 0.72 | 0.55 | 6.0 | −0.2 | 0.81 [0.810, 0.812] | 0.76 [0.755, 0.757] | 4.1 [4.05, 4.07] | −1.3 [−1.31, −1.28] | 0.81 [0.810, 0.812] | 0.75 [0.744, 0.747] | 4.1 [4.05, 4.07] | −1.3 [−1.31, −1.28] |
| C | 0.71 | 0.58 | 6.0 | −2.2 | 0.79 [0.793, 0.797] | 0.77 [0.763, 0.767] | 4.4 [4.35, 4.38] | −0.4 [−0.44, −0.41] | 0.74 [0.740, 0.744] | 0.70 [0.693, 0.698] | 5.0 [4.94, 4.97] | −0.5 [−0.52, −0.48] |
| $\rho_b$ (g·cm$^{-3}$) | | | | | | | | | | | | |
| A | **0.48** | **0.46** | **0.15** | **0.00** | 0.28 [0.276, 0.281] | 0.16 [0.151, 0.160] | **0.15** [0.146, 0.147] | **0.00** [0.002, 0.003] | 0.27 [0.268, 0.273] | 0.13 [0.128, 0.136] | **0.15** [0.147, 0.148] | **0.01** [0.014, 0.015] |
| B | **0.47** | **0.29** | **0.21** | −0.04 | 0.33 [0.333, 0.336] | 0.27 [0.272, 0.276] | **0.18** [0.181, 0.181] | **0.01** [0.004, 0.005] | 0.32 [0.315, 0.319] | 0.26 [0.256, 0.260] | **0.18** [0.183, 0.184] | **0.01** [0.005, 0.006] |
| C | 0.40 | 0.12 | 0.28 | −0.08 | **0.53** [0.525, 0.529] | **0.48** [0.476, 0.480] | **0.18** [0.179, 0.179] | **0.00** [−0.001, 0.000] | 0.52 [0.517, 0.521] | 0.47 [0.470, 0.474] | **0.18** [0.180, 0.181] | **0.00** [−0.001, 0.000] |
| $K_{sat}$ (cm·h$^{-1}$) | | | | | | | | | | | | |
| A | 0.00 | −0.28 | 50.0 | −23.5 | **0.15** [0.146, 0.150] | **−0.10** [−0.100, −0.099] | 30.7 [30.6, 30.8] | −12.5 [−12.6, −12.4] | 0.13 [0.129, 0.133] | **−0.10** [−0.107, −0.101] | 31.2 [31.0, 31.3] | −12.7 [−12.8, −12.6] |
| B | 0.05 | −0.14 | 34.3 | −13.8 | **0.42** [0.418, 0.423] | **0.29** [0.287, 0.292] | 22.9 [22.8, 23.1] | −6.7 [−6.8, −6.6] | 0.37 [0.365, 0.370] | 0.23 [0.232, 0.238] | 23.5 [23.4, 23.6] | **−6.7** [−6.8, −6.6] |
| C | 0.11 | −15.51 | 25.5 | −3.3 | 0.38 [0.381, 0.386] | −0.08 [−0.113, −0.046] | **12.3** [12.2, 12.4] | −2.6 [−2.7, −2.6] | **0.38** [0.377, 0.383] | −0.15 [−0.185, −0.106] | 12.5 [12.4, 12.6] | **−2.6** [−2.7, −2.6] |

Accuracy assessment plots of the PTFs developed *to estimate* $\theta_{33}$ are presented in Figure 4a. PTFs'S&R.2006 for A and C horizons had a lower $R^2$ than both the SFR- and the CCA-derived PTFs. However, they also had the best $R^2$ for the B horizon. Nevertheless, PTFs'S&R.2006 were more erroneous in terms of RMSE and bias. Estimation quality for $\theta_{33}$ was very similar with both the SFR and the CCA methods ($R^2$ values ranging from 0.53 to 0.70). In terms of NSE, the performance of PTFs'S&R.2006 was unsatisfactory with a negative value for the A horizon, satisfactory for the B horizon (moderately), and unsatisfactory for the C horizon (weak). The SFR- and the CCA-derived PTFs performed equally well with a moderate NSE for the A horizon and a good NSE for the B and C horizons.

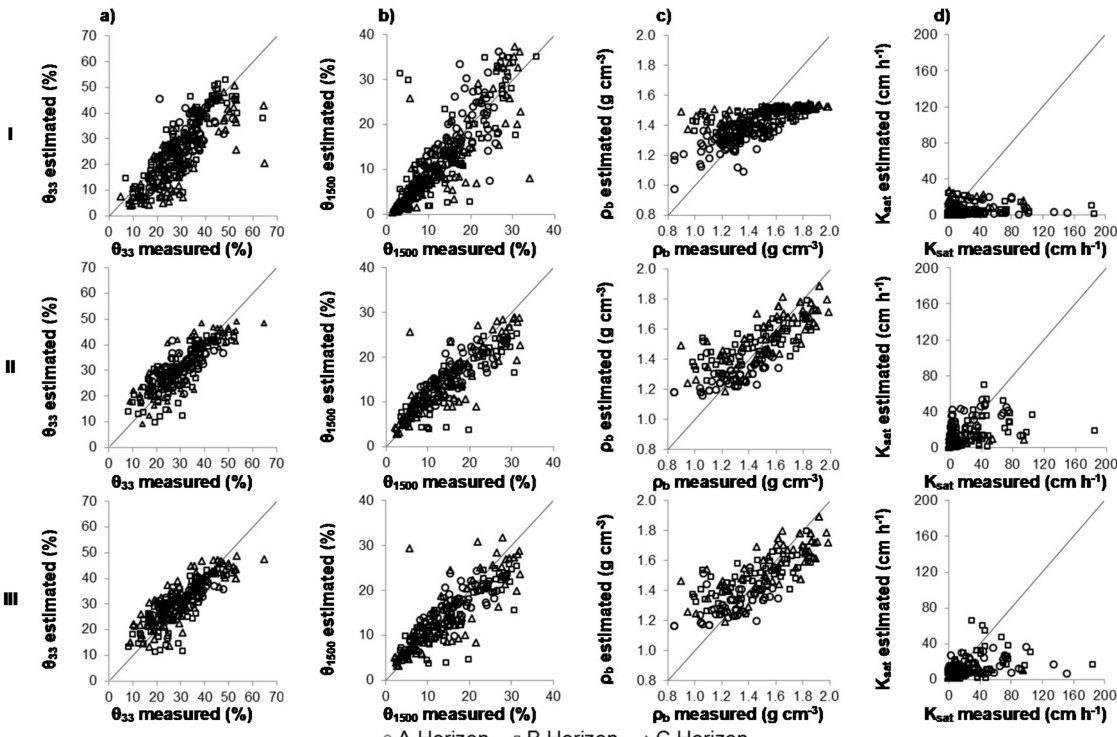

**Figure 4.** Accuracy assessment of Saxton and Rawls's PTFs (**I**) and PTFs developed using SFR (**II**) and CCA (**III**): (**a**) field capacity ($\theta_{33}$); (**b**) permanent wilting point ($\theta_{1500}$); (**c**) bulk density ($\rho_b$); (**d**) saturated hydraulic conductivity ($K_{sat}$).

Accuracy assessment plots of the PTFs developed *to estimate $\theta_{1500}$* are illustrated in Figure 4b. The same pattern as that observed with PTFs'S&R.2006 used to predict $\theta_{33}$ was found with $\theta_{1500}$. In comparison with $\theta_{33}$, the performance of $\theta_{1500}$ PTFs was slightly higher. The $R^2$ values were good, especially for the B and C horizons. In terms of NSE, the performance of PTFs'S&R.2006 was unsatisfactory for the A horizon (weak) and satisfactory for the B and C horizons (moderate). The SFR-derived PTFs outperformed the CCA-derived PTFs; however, for both methods, the NSE was qualified as good for all horizons.

Accuracy assessment plots of the PTFs developed *to estimate $\rho_b$* are illustrated in Figure 4c. PTFs'S&R.2006 gave the best $\rho_b$ prediction for the A horizon, with the highest $R^2$ and lowest RMSE (0.48 and 0.15 g·cm$^{-3}$). For this horizon, the SFR- and CCA-derived PTFs led to the same $R^2$ and RMSE values (0.28 and 0.15 g·cm$^{-3}$, respectively). Both methods also generated the same $R^2$ and RMSE values at the B and C horizons. In terms of NSE, the performance of PTFs'S&R.2006 was satisfactory (moderate) for the A horizon and unsatisfactory (weak) for the B and C horizons. The SFR- and the CCA-derived PTFs performed equally well with a weak NSE for the A and B horizons and a moderate NSE for the C horizon.

Accuracy assessment plots of the PTFs generated *to estimate $K_{sat}$* are illustrated in Figure 4d. At the A horizon, the SFR method performed better than the CCA method. In comparison to PTFs'S&R.2006 and the CCA-derived PTF, the SFR-derived PTF had a higher $R^2$ and a lower RMSE. However, its estimation quality remained weak ($R^2$ of 0.15 and RMSE of 30.7 cm·h$^{-1}$). The estimation quality of PTFs'S&R.2006 was poor for the B horizon. For this horizon, both CCA- and SFR-derived PTFs were slightly more accurate ($R^2$ of 0.42 and RMSE of 22.9 cm·h$^{-1}$ for SFR-PTFs). At the C horizon, results were similar to those of the B horizon, but with lower RMSE (12.3 cm·h$^{-1}$). In terms of NSE, PTFs'S&R.2006 were unsatisfactory for all horizons (negative values). The same conclusion was noted for both SFR- and CCA-derived PTFs, but with weak positive values for the B horizon. The performance remained unsatisfactory. The SFR-derived PTFs slightly outperformed the

CCA-derived PTFs; however, for both methods, the NSE was qualified as unacceptable for the A and C horizons and weak for the B horizon.

## 4. Discussion

The above results demonstrate the potential of the primary soil properties (clay, silt, OC, and CF percentages) to estimate the secondary soil properties ($\theta_{1500}$, $\theta_{33}$, $\rho_b$, and $K_{sat}$) for different horizons (A, B, and C) with varying accuracy rates. The hypothesis tested in this study was the ability of locally trained PTFs, using SFR and CCA algorithms, to produce more accurate estimates than the PTFs'S&R.2006 that were trained with global soil data. As expected, our results highlighted that, in most cases, locally trained PTFs achieved the best accuracy in estimating secondary soil properties.

*In the case of $\theta_{33}$*, PTFs'S&R.2006 results were systematically underestimated. This underestimation increased in the deeper horizons (soil depth), which is unsurprising, since the PTFs'S&R.2006 were developed using A horizon soil samples. Both the SFR- and CCA-derived PTFs had moderate performances for the A horizon and good performances for the B and C horizons. Pollacco [64] evaluated the performance of eight different PTFs to predict $\theta_{33}$. The RMSE values for these PTFs ranged from 5.7% to 11.1%, whereas they ranged from 4.3% to 7.4% for the PTFs developed in the present study, which is considered acceptable. These results are comparable to the $\theta_{33}$ accuracy assessment results found in the literature [2,28,65].

*In the case of $\theta_{1500}$*, the PTFs'S&R.2006 were again less accurate with a lower $R^2$ and higher RMSE and bias than the SFR- and CCA-derived PTFs for the A horizon. Methods developed using SFR and CCA showed similar performance. The SFR method produced slightly higher $R^2$ and lower RMSE and bias at the A and C horizons. The performance of the two methods was identical at the B horizon. In fact, the clay percentage was strongly correlated with $\theta_{1500}$ (Figure 3). The reason is that the SFR method selected clay as a predictor, while the CCA method selected the canonical variable to which clay was the major contributor. Both SFR- and CCA-derived PTFs were satisfactory (good) for all horizons. Again, RMSE values obtained in this study (ranging from 3.2% to 5.0%) outperformed the RMSE values obtained by Pollacco [64] (ranging from 4.7% to 7.5%). These results are acceptable and similar to the $\theta_{1500}$ results found in the literature [2,28,65].

*In the case of $\rho_b$ estimation*, PTFs'S&R.2006 were well adapted to the A horizon. This is likely because the PTFs'S&R.2006 use volumetric water content at saturation (obtained by a PTF) to generate normal density, which is then used with CF to predict $\rho_b$, and they were originally calibrated using the A horizon data. At the B horizon, in terms of estimation errors, our PTFs were less erroneous than the PTFs'S&R.2006. The latter produced the highest $R^2$ (0.47) at the B horizon, but it also showed an important RMSE value (0.21 g·cm$^{-3}$). Both the SFR- and CCA-derived PTFs outperformed PTFs'S&R.2006 at the C horizon in terms of $R^2$ and RMSE (0.53 vs. 0.40 and 0.18 vs. 0.28 g·cm$^{-3}$). A comparison with PTFs available in the literature [2,66,67] suggests that RMSE should range between 0.13 and 0.23 g·cm$^{-3}$, which was the case.

*In the case of $K_{sat}$*, the estimation quality was poor for all horizons. The SFR method slightly outperformed the CAA method at the B horizon, and both methods performed poorly for the A and C horizons yet showed higher accuracy than PTFs'S&R.2006 estimates. In fact, the soil of the A horizon is frequently disturbed by tillage, plant root penetration, and field alterations that modify soil structure. This variation in soil structure could explain the high variability observed in saturated soil hydraulic conductivity (Table 2), which is not explained by soil texture and OC alone. In addition, $K_{sat}$ and CF showed great variability; consequently, averages were less representative for these properties than for other soil properties. Jorda et al. [26] found that the most influential predictor for $K_{sat}$ was land use. They noted a difference between samples in conventional agricultural sites and non-tilled sites. These results demonstrate the importance of soil structure to $K_{sat}$ estimations. The lack of selected predictors that relate to soil structure might explain the

difficulty to estimate $K_{sat}$. Prediction of this physical property could, however, be improved by adding morphological predictors such as soil structure and drainage information.

Overall, the SFR-derived PTFs were equal to or more accurate than the CCA-derived PTFs and were more accurate than the PTFs'S&R.2006. In fact, the major difference between CCA and SFR methods is that CCA always uses all available predictors to develop its canonical variables, whereas the SFR method uses a selection strategy to determine explanatory predictors. The use of all predictors may introduce noise, which explains the observed difference in performance between the two developed PTFs. Even in the few cases where the CCA-derived PTFs outperformed the SFR-derived PTFs ($\theta_{33}$ for horizon C, for example (Table 8)), we still recommend using the SFR-derived PTF because of the small significant difference in results and its easier execution.

## 5. Conclusions

In this study, four secondary soil properties—bulk density ($\rho_b$), saturated hydraulic conductivity ($K_{sat}$), and volumetric water content ($\theta$) measured at two matric potentials, $-33$ kPa (field capacity ($\theta_{33}$)) and $-1500$ kPa (permanent wilting point ($\theta_{1500}$))—were estimated for A, B, and C horizons for agricultural areas of southern Quebec, Canada. Estimates were performed using existing functions from Saxton and Rawls's PTFs (2006) and new PTFs trained using the stepwise forward regression (SFR) and canonical correlation analysis (CCA) algorithms. Primary soil properties (clay, silt, organic carbon, and coarse fragment percentages) were used as inputs for estimating the secondary soil properties. All PTFs (equations are available in Appendix A) were assessed using the cross-validation technique from which the $R^2$, Nash–Sutcliffe efficiency (NSE), root-mean-square error (RMSE), and bias were generated. Except for $\rho_b$ for A and B horizons that showed higher accuracy in terms of NSE and $R^2$ and equal accuracy in terms of RMSE and bias, all the other physical secondary soil properties estimated using either SFR- or CCA-derived PTFs were more accurate, particularly for $\theta_{33}$ and $\theta_{1500}$. According to the NSE index, $\theta_{1500}$ showed the best performance (qualified as good), followed by $\theta_{33}$ (qualified as moderate to good) for the different horizons. The NSE index for $\rho_b$ was qualified as low, while the NSE index for $K_{sat}$ was qualified as unacceptable to low for the best performances for the different horizons. It is, thus, recommended to retrain the last two soil properties using other morphological predictors such as soil structure and drainage information before considering their use. Overall, the SFR method showed equal to or better performance than the CCA method.

**Author Contributions:** Conceptualization, S.P. and K.C.; methodology, S.P. and K.C.; validation, S.P. and K.C.; formal analysis, S.P. and K.C.; resources, K.C.; data curation, S.P. and A.N.C.; writing—original draft preparation, S.P.; writing—review and editing, A.E.A., S.P. and K.C.; visualization, A.E.A.; supervision, K.C.; project administration, S.P.; funding acquisition, K.C. All authors have read and agreed to the published version of the manuscript.

**Funding:** This study was funded by the SAGES program of Agriculture and Agri-Food Canada.

**Institutional Review Board Statement:** Not applicable.

**Informed Consent Statement:** Not applicable.

**Data Availability Statement:** All data collected, preprocessed, processed, or analyzed during this study are included in this work.

**Acknowledgments:** The authors thank the Pedology and Precision Agriculture Laboratories Staff of Agriculture and Agri-Food Canada. The authors are also especially grateful to André Martin, Claude Lévesque, and Mario Deschênes for their chemical and physical soil analyses and to Luc Lamontagne for his soil survey expertise. Our gratitude also goes to Michel C. Nolin and Gaétan Bourgeois for their tremendous contribution to this project.

**Conflicts of Interest:** The authors declare no conflict of interest.

## Appendix A. Canonical Correlation Analysis and Stepwise Forward Regression PTFs Developed for A, B, and C Horizons

**Table A1.** PTFs developed for the A horizon.

| Stepwise Forward Regression Method | $R^2$ | NSE | RMSE | Bias |
|---|---|---|---|---|
| $\theta_{33} = 1.8373 + 0.1903 \cdot Si + 1.5476 \cdot C^{0.4072} + 15.0148 \cdot OC^{0.2750}$ | 0.54 | 0.47 | 4.3 | 0.4 |
| $\theta_{1500} = 0.1266 \cdot Si + 3.4684 \cdot C^{0.4072} + 7.8006 \cdot OC^{0.2750} - 10.0548$ | 0.68 | 0.67 | 3.2 | 0.1 |
| $\rho_b = -0.0478 \cdot C^{0.4072} - 0.5903 \cdot OC^{0.2750} + 2.1906$ | 0.28 | 0.16 | 0.15 | 0.00 |
| $K_{sat} = e^{0.1309 + 1.6519 \cdot OC^{0.2750} + 0.3674 \cdot \ln(CF+1)} - 1$ | 0.15 | −0.10 | 30.7 | −12.5 |
| **Canonical Correlation Analysis Method** | | | | |
| $\theta_{33} = 32.3500 + 5.3145 \cdot U_1$ | 0.53 | 0.47 | 4.4 | 0.4 |
| $\theta_{1500} = 15.7784 + 4.6768 \cdot U_1 + 1.3632 \cdot U_2$ | 0.66 | 0.61 | 3.33 | 0.3 |
| $\rho_b = 1.2982 - 0.1266 \cdot U_1 + 0.0546 \cdot U_2$ | 0.27 | 0.13 | 0.15 | 0.01 |
| $K_{sat} = e^{2.4895 - 0.5768 \cdot U_3} - 1$ | 0.13 | −0.10 | 31.2 | −12.7 |
| $U_1 = 0.0217 \cdot Si + 0.5238 \cdot \left(C^{0.4072} - 1\right) + 2.8083 \cdot \left(OC^{0.2750} - 1\right) - 0.0.0673 \cdot \ln(CF+1) - 2.6251$ | | | | |
| $U_2 = 0.0118 \cdot Si + 0.8132 \cdot \left(C^{0.4072} - 1\right) - 3.7373 \cdot \left(OC^{0.2750} - 1\right) + 0.2768 \cdot \ln(CF+1) - 1.6776$ | | | | |
| $U_3 = 0.0382 \cdot Si - 0.4248 \cdot \left(C^{0.4072} - 1\right) - 1.3209 \cdot \left(OC^{0.2750} - 1\right) - 0.6684 \cdot \ln(CF+1) + 0.5212$ | | | | |

**Table A2.** PTFs developed for the B Horizon.

| Stepwise Forward Regression Method | $R^2$ | NSE | RMSE | Bias |
|---|---|---|---|---|
| $\theta_{33} = 0.0943 \cdot Si + 43.8066 \cdot C^{0.1248} - 13.0937 \cdot OC^{-0.0874} + 1.6381 \cdot \ln(CF+1) - 20.0657$ | 0.68 | 0.63 | 5.9 | 0.8 |
| $\theta_{1500} = e^{\frac{\ln\left(-0.0643 + 0.0019 \cdot Si + 1.2369 \cdot C^{0.1248}\right)}{0.2258}}$ | 0.81 | 0.76 | 4.1 | 1.3 |
| $\rho_b = e^{\frac{\ln\left(-0.6911 + 0.0165 \cdot Si - 2.2053 \cdot C^{0.1248} + 5.3959 \cdot OC^{-0.0874}\right)}{2.4488}}$ | 0.33 | 0.27 | 0.18 | 0.01 |
| $K_{sat} = e^{9.8823 - 0.0340 \cdot Si - 6.4758 \cdot OC^{-0.0874} + 0.2583 \cdot \ln(CF+1)} - 1$ | 0.42 | 0.29 | 22.9 | 6.7 |
| **Canonical Correlation Analysis Method** | | | | |
| $\theta_{33} = 28.7736 - 9.4011 \cdot U_1$ | 0.68 | 0.63 | 5.9 | 0.8 |
| $\theta_{1500} = e^{\frac{\ln\left(1.7521 - 0.2199 \cdot U_1\right)}{0.2258}}$ | 0.81 | 0.75 | 4.1 | −1.3 |
| $\rho_b = e^{\frac{\ln\left(2.7343 + 0.3196 \cdot U_1 - 0.5772 \cdot U_2\right)}{2.4488}}$ | 0.32 | 0.26 | 0.18 | 0.01 |
| $K_{sat} = e^{\left(1.8381 + 0.4429 \cdot U_1 + 0.8850 \cdot U_2\right)} - 1$ | 0.37 | 0.23 | 23.5 | −6.7 |
| $U_1 = -0.0101 \cdot Si - 4.9887 \cdot \left(C^{0.1248} - 1\right) + 0.7156 \cdot \left(OC^{-0.0874} - 1\right) + 0.1170 \cdot \ln(CF+1) + 2.2069$ | | | | |
| $U_2 = -0.0329 \cdot Si + 1.4155 \cdot \left(C^{0.1248} - 1\right) - 8.3523 \cdot \left(OC^{-0.0874} - 1\right) + 0.1326 \cdot \ln(CF+1) + 1.2917$ | | | | |

**Table A3.** PTFs developed for the C Horizon.

| Stepwise Forward Regression Method | $R^2$ | NSE | RMSE | Bias |
|---|---|---|---|---|
| $\theta_{33} = 0.1815 \cdot Si + 24.1479 \cdot C^{0.1659} + 25.2204 \cdot OC^{0.1664} - 2.6145 \cdot \ln(CF+1) - 28.1507$ | 0.69 | 0.65 | 7.4 | 0.5 |
| $\theta_{1500} = e^{\frac{\ln\left(0.0025 \cdot Si + 0.8846 \cdot C^{0.1659} + 0.2567\right)}{0.2206}}$ | 0.79 | 0.77 | 4.4 | 0.4 |
| $\rho_b = e^{\frac{\ln\left(0.0171 \cdot Si - 1.0133 \cdot C^{0.1659} - 5.4107 \cdot OC^{0.1664} + 0.2408 \cdot \ln(CF+1) + 7.3751\right)}{2.2516}}$ | 0.53 | 0.48 | 0.18 | 0.00 |
| $K_{sat} = e^{-0.0374 \cdot Si - 0.9415 \cdot C^{0.1659} + 2.8791 \cdot OC^{0.1664} + 1.7888} - 1$ | 0.42 | −0.08 | 12.3 | −2.6 |
| **Canonical Correlation Analysis Method** | | | | |
| $\theta_{33} = 29.5832 + 10.6868 \cdot U_1$ | 0.70 | 0.66 | 7.3 | 0.5 |
| $\theta_{1500} = e^{\frac{\ln\left(1.6935 + 0.2545 \cdot U_1\right)}{0.2206}}$ | 0.74 | 0.70 | 5.0 | −0.5 |
| $\rho_b = e^{\frac{\ln\left(2.8777 - 0.5208 \cdot U_1 + 0.5467 \cdot U_2\right)}{2.2516}}$ | 0.52 | 0.47 | 0.18 | 0.00 |
| $K_{sat} = e^{\left(1.1059 - 0.4499 \cdot U_1 - 0.5372 \cdot U_2 + 0.2169 \cdot U_3\right)} - 1$ | 0.38 | −0.15 | 12.5 | −2.6 |
| $U_1 = 0.0103 \cdot Si + 2.9005 \cdot \left(C^{0.1659} - 1\right) + 1.8130 \cdot \left(OC^{0.1664} - 1\right) - 0.1148 \cdot \ln(CF+1) - 1.2149$ | | | | |
| $U_2 = 0.0458 \cdot Si + 0.4749 \cdot \left(C^{0.1659} - 1\right) - 7.8326 \cdot \left(OC^{0.1664} - 1\right) + 0.2432 \cdot \ln(CF+1) - 4.4052$ | | | | |
| $U_3 = -0.0372 \cdot Si + 3.4037 \cdot \left(C^{0.1659} - 1\right) - 2.7869 \cdot \left(OC^{0.1664} - 1\right) + 0.6708 \cdot \ln(CF+1) - 2.1718$ | | | | |

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
