# Peer review of "Development of Pedotransfer Functions to Predict Soil Physical Properties in Southern Quebec (Canada)"

_agronomy, doi:10.3390/agronomy12020526_

Round 1
Reviewer 1 Report
Authors had revised the manuscript according to my comments. Good job.
Author Response
To Reviewer 1

Reviewer 2 Report
This paper presents a study of applying SFR and CCA to identify the Pedotransfer functions that are predictive functions to estimate soil properties. The study compares the results of already developed pedotransfer functions by Saxton and Rawls, 2006.
The paper is quite OK, but nothing new is in this paper. It is just an application paper. It is only a study of two very well-known methods to estimate different functions.
Abstract and Keywords: I suggest removing the first two keywords as those two are already in the title.
Since authors developed the paper, and now it is well-shaped; I do not have any reservations.
Author Response
To Reviewer 2

This manuscript is a resubmission of an earlier submission. The following is a list of the peer review reports and author responses from that submission.
Round 1
Reviewer 1 Report
This paper presents a study of applying SFR and CCA to identify the Pedotransfer functions that are predictive functions to estimate soil properties. The study compares the results of already developed pedotransfer functions by Saxton and Rawls, 2006.
The paper is quite OK, but nothing new is in this paper. It is just an application paper. It is only a study of two very-well known methods to estimate different function.
Abstract and Keywords
To improve the paper, the authors need to modify the abstract, it is very confuse and it is not clear (line 22-24). When you read the paper, you can understand the abstract, but the abstract does not give a clear summary of the paper.
Abstract has some typo errors too (line 15 – CAA)
This manuscript mentioned nine keywords, seems too much, however depends on the Journal’s guidelines.
Introduction
Need improvement in this section. Better to add more references for different methods of PTF development (for traditional methods (ie regression) and for machine leaning techniques).
Also need to brief about Saxton and Rawls, 2006 PTFs
M&M
Some references not found in this list.
Need some more references for the section 2.3.3
Figure 4 shows high variability, and in turn high standard error, of the simulated values compared to the observed ones. This will have an impact on the reliability of the estimates
R&D
Need references (line 251-253).
Some references not found (line 222 – “Error! Reference source not found”
The conclusion that the parameters estimated with a good accuracy is questionable, as these estimated values can be within +-50% of the observed values!! The use of these simulated values, when they are away from the “correct” ones, can produce in unreliable results. It is recommended that the authors should be careful in saying this!!
Table 8 not clear
Conclusions
Need to rewrite by removing unnecessary wage opinions and draw solid conclusions using your results.
Author Response
Responses are in the word file
Reviewer 2 Report
This study was designed to develop a new set of PTFs that are well adapted to the pedoclimatic conditions of the agricultural area of southern Quebec. The study was well designed and carefully conducted, and the results are not addressing important questions about predicting soil physical properties in southern Quebec. However, the quality of the manuscript was relatively lower.
I also have major concerns regarding the publication for this manuscript. My first major concern is that authors did not give any hypothesis. Based on the summary of previous studies, authors should clearly present the aims and hypothesis of this study and discuss hypothesis in the “Discussion”.
My second major concern is that the discussion section of the current manuscript is not written, and this part needs to separate from results. Authors could setup 2-3 subtitles in the discussion section to discuss hypothesis or new findings.
My third major concern is that “the conclusion” in the manuscript is not the real conclusion. Authors should simplify this part to 200-300 words, and revealed implications about the findings of this manuscript.
Author Response
Responses are in the word file
